# Adjuvant Chemotherapy in Older Patients with Gastric Cancer: A Population-Based Cohort Study

**DOI:** 10.3390/cancers15153768

**Published:** 2023-07-25

**Authors:** Wing-Lok Chan, Xiaodong Liu, Carlos King-Ho Wong, Michael Siu-Nam Wong, Ian Yu-Hong Wong, Ka-On Lam, Bryan Ho-Kwan Yun, Emina Edith Cheung, Rosa Pui-Ying Tse, Fion Chan, Simon Law, Dora Kwong

**Affiliations:** 1Department of Clinical Oncology, School of Clinical Medicine, Li Ka Shing Faculty of Medicine, The University of Hong Kong, Hong Kong SAR, China; xmichael@hku.hk (M.S.-N.W.); lamkaon@hku.hk (K.-O.L.); bryanyhk@hku.hk (B.H.-K.Y.); cheunge1@hku.hk (E.E.C.); dlwkwong@hku.hk (D.K.); 2Department of Surgery, School of Clinical Medicine, Li Ka Shing Faculty of Medicine, The University of Hong Kong, Hong Kong SAR, China; u3006667@connect.hku.hk (X.L.); iyhwong@hku.hk (I.Y.-H.W.); syklaw@hku.hk (S.L.); 3Department of Pharmacology and Pharmacy, Li Ka Shing Faculty of Medicine, The University of Hong Kong, Hong Kong SAR, China; carlosho@hku.hk; 4Department of Clinical Oncology, Queen Mary Hospital, Hong Kong SAR, China; tpy804@ha.org.hk; 5Department of Surgery, Queen Mary Hospital, Hong Kong SAR, China; fsychan@hku.hk

**Keywords:** adjuvant chemotherapy, gastric cancer, older patients, geriatric oncology, comorbidity

## Abstract

**Simple Summary:**

This real-world population-based cohort study evaluated the efficacy of adjuvant chemotherapy in older patients (aged ≥65 years old) after D2-gastrectomy. Adjuvant chemotherapy was associated with better overall survival and cancer-specific survival than surgery alone. There was no significant difference in survival benefit between those patients who received monotherapy or doublets chemotherapy.

**Abstract:**

(1) Background: The effectiveness of adjuvant chemotherapy in older patients with gastric cancer after D2-gastrectomy is unclear. This study investigated the efficacy of adjuvant chemotherapy in elderly patients with stage II/III gastric cancer. (2) Methods: A real-world population-based retrospective cohort of patients aged ≥65 with stage II/III gastric cancer (n = 2616; median age: 73.5; 12.2% aged >80 years) treated between 1 January 1997 and 31 December 2020 were included. All data was retrieved from the Hong Kong Hospital Authority Clinical Management System (CMS). Clinical characteristics of those patients with and without adjuvant chemotherapy treatment were balanced after propensity score matching (PSM). In total, 732 patients treated with adjuvant chemotherapy were matched with 732 patients treated with surgery alone. Hazard ratios (HRs) estimated via Cox proportional hazards regression models were used to compare the overall survival (OS) and cancer-specific survival (CSS) of the two patient groups. (3) Results: Adjuvant chemotherapy was associated with better OS (37 vs. 25 months; HR: 0.80; 95% CI: 0.75–0.84; *p* < 0.001) than surgery alone. The OS benefit was observed in both the 65–80 (44 vs. 27 months; HR: 0.79; 95% CI: 0.74–0.84; *p* < 0.001) and >80 (14 vs. 11 months; HR: 0.82; 95% CI: 0.71–0.96; *p* < 0.001) age groups. A better CSS was observed in patients who received adjuvant chemotherapy than those who only had surgery (5-year CSS: 64.1% vs. 61.1%, HR: 0.85; 95% CI: 0.79–0.93; *p* < 0.001). (4) Conclusions: adjuvant chemotherapy significantly improved OS and CSS in older patients with stage II/III gastric cancer.

## 1. Introduction

Gastric cancer is the fifth most common malignancy and fourth leading cause of cancer death worldwide [1]. Its incidence and mortality are highly variable by region and dependent on diet and the prevalence of Helicobacter pylori infection [2,3]. It has a high prevalence in East Asia, especially Japan, Korea, and Southern China. Gastric cancer is more prevalent in the older populations. According to the Surveillance, Epidemiology, and End Results (SEER) database, more than 60% of gastric cancer cases develop in patients over 65 years old, and about one-third of patients are over 75 years old [4].

Currently, gastrectomy with extended lymph node dissection (D2-dissection) is the standard method of care provided to patients with localised gastric cancer in Asia [5,6,7,8]. However, after curative resection (R0), the prognosis remains poor, and there is a high risk of disease recurrence, with exact risk ranging from 30 to 60% [9,10]. To reduce the risk of relapse and improve survival, multimodality treatments, such as perioperative chemotherapy or adjuvant chemotherapy/chemoradiotherapy, have been extensively examined over the past two decades [11,12,13,14,15,16,17]. Based on the promising results of the two landmark randomised controlled trials (RCTs), which are known as CLASSIC and ACT-GC, post-operative adjuvant chemotherapy is usually prescribed for stage II/III gastric cancer treatment in Asia [18,19].

As the world’s population is aging, the incidence of gastric cancer is increasing, and the management of gastric cancer in older populations has become more challenging. Older patients usually have more comorbidities, shorter overall survival (OS), less frequently operations, and a higher risk of complications [20]. Moreover, older patients are often under-represented in clinical trials. The older patients enrolled in clinical trials are usually fitter and have fewer comorbidities than those patients in routine clinical settings. The recommendations in clinical guidelines are often suited to younger adults. Therefore, the effectiveness of adjuvant chemotherapy for older patients should be cautiously assessed to avoid over- or under-treatment. This large population-based cohort study aims to investigate the effectiveness of adjuvant chemotherapy in elderly patients with stage II/III gastric cancers after D2 gastrectomy.

## 2. Materials and Methods

### 2.1. Patient Eligibility and Data Collection

In this real-world population-based cohort study, demographic and clinical data of patients aged ≥65 with stage II/III gastric cancer treated between 1 January 1997 and 31 December 2020 were collected from the territory-wide prospectively coded database (the Hong Kong Hospital Authority Clinical Management System [CMS]). The CMS database is the largest cancer database in Hong Kong, and it includes approximately 90% of cancer cases. Data collected from the database contained hospitalisation records, histories of medical dispensing, laboratory test results, treatment procedures, comorbidities, and mortalities. The study was conducted in accordance with the Declaration of Helsinki, and approved by the Institutional Review Board of the Hong Kong West Cluster Institutional Review Board (protocol code UW 18-506 and approved on 30 December 2021).

Eligible patients included those who were aged ≥65 at diagnosis, had a diagnosis of stage II/III gastric cancer, underwent gastrectomy with D2-dissection, and had a follow-up period of at least one month. Exclusion criteria included patients who were aged <65 years old, had stage I or IV gastric cancer, previously received neo-adjuvant chemotherapy or radiotherapy, experienced a relapse of gastric cancer within 6 months, died within one month of diagnosis, or had a follow-up period of less than one month since starting chemotherapy.

Using the medical records in the dataset, the cohort was categorised into two groups: adjuvant chemotherapy and surgery alone. The adjuvant chemotherapy group included patients who received chemotherapy within 6 months of undergoing a gastrectomy. The commonly used chemotherapy agents included 5-fluorouracil, S-1, capecitabine, oxaliplatin, cisplatin, docetaxel, and epirubicin. The surgery alone group included patients who underwent a radical gastrectomy, but did not receive adjuvant chemotherapy. The index date of the subjects was defined as the date of gastric cancer diagnosis. All patients had a follow-up period lasting until the date of death or the data cut-off date (31 December 2020), whichever came first.

### 2.2. Baseline Covariates

Baseline covariates included demographic characteristics, biological parameters, history of comorbidities, and concomitant treatments. Of these factors, biological parameters, including complete blood count (white blood cell [WBC], hemoglobin [Hb], platelet, neutrophil, and lymphocyte), liver function tests (serum albumin, total bilirubin, alkaline phosphatase [ALP], alanine aminotransferase [ALT], and aspartate aminotransferase [AST]), and renal function tests (serum urea and estimated glomerular filtration rate [eGFR]), were collected. The history of comorbidities retrieved from the database included coronary heart disease, congestive heart failure, stroke, atrial fibrillation, diabetes, hypertension, renal impairment, and liver disease. The Charlson Comorbidity Index (CCI) scores were calculated. The number of chemotherapy and regimens used were also analysed.

### 2.3. Outcomes

The study outcome was determining OS and cancer-specific survival (CSS). OS was defined as the duration of the period extending from the date of diagnosis to death or the last follow-up date, with no restriction placed on the cause of death. CSS was defined as the duration of the period from the date of diagnosis to death or the last follow-up date, with the cause of death being restricted to gastric cancer.

Severe adverse events, including grade 3/4 anemia, neutropenia, thrombocytopenia, and severe infections (febrile neutropenia, fever or pneumonia, or urinary tract infection that required intravenous antibiotics), were collected for the adjuvant chemotherapy group according to the NCI Common Terminology Criteria for Adverse Events (CTCAE) v4.0.

### 2.4. Statistical Analyses

Multiple imputation by chained equations (MICE) was performed to address missing baseline data. Next, the propensity score matching (PSM) in a ratio of 1:1 was used to match patients who received adjuvant chemotherapy with those who received surgery alone. Patients with adjuvant chemotherapy and those who received surgery alone were matched for age, gender, comorbidities, and biological parameters, including WBC, Hb level, eGFR, and albumin, which were collected around 4 weeks after the operation (and before the start of chemotherapy in those who received adjuvant chemotherapy).

Cox proportional hazards regression models were performed to assess the relative risk of survival outcomes between the adjuvant chemotherapy and surgery alone groups. The results obtained through the estimations were shown as hazard ratios (HR) and their 95% confidence intervals (CI). Kaplan–Meier curves, in combination with the log-rank test, were used to compare the differences between OS and CSS between patients who received and did not receive adjuvant chemotherapy. The median OS and 5-year survival rates were also calculated.

Subgroup analyses were performed based on age at diagnosis (65–80, >80 years), sex (female, male), history of diabetes, CCI (<8, ≥8), albumin level (<30, 30–50 g/L), and eGFR level (with cutoff using 60 mL/min/1.73 m^2^ and 90 mL/min/1.73 m^2^). Multivariable logistic regression models were used to explore the risk factors associated with OS and CSS in older patients with gastric cancer.

A sensitivity analysis was conducted to test the robustness of the results by including all characteristics, comorbidities, and biological parameters in the PSM. A multivariable logistic regression model was used to explore the risk factors associated with OS in older patients with gastric cancer. In addition, analysis was performed to compare the survival outcomes between patients who received either monotherapy or doublet chemotherapies.

All statistical analyses were performed using Stata version 16.0 (Stata Corp LP, College Station, TX, USA). A two-tailed *p*-value of <0.05 was considered statistically significant.

## 3. Results

Data belonging to a total of 14,694 patients diagnosed with adenocarcinoma of the stomach were retrieved from the database. After excluding ineligible subjects, 2616 patients aged ≥65 years who performed D2-gastrectomy for stage II/III gastric cancer were identified. Figure 1 shows the flowchart of the patient selection process.

Of the identified patients, 1706 (65.2%) were men and 910 (34.8%) were women; the mean (SD) age was 75.8 (±6.8) years old, and 648 (24.8%) patients were aged 80 years or above. A total of 923 (35.3%) patients received adjuvant chemotherapy, while 1693 (64.7%) patients did not receive this treatment. After PSM, a total of 732 patients who received adjuvant chemotherapy were matched with another 732 patients who underwent surgery alone. The baseline characteristics of patients were well balanced between two groups after PSM. Table 1 shows the baseline characteristics of patients after PSM.

Compared to surgery alone, adjuvant chemotherapy was associated with a statistically significant improvement in OS (median OS: 37 vs. 25 months; 5-year OS rate: 44.0% vs. 35.8%; HR: 0.80; 95% CI: 0.75–0.84, *p* < 0.001). The improvement in OS was significant in both the 65–80 (44 vs. 27 months, HR: 0.79, 95% CI: 0.74–0.84, *p* < 0.001) and >80 (14 vs. 11 months, HR: 0.82, 95% CI: 0.71–0.96, *p* < 0.001) subgroups. A better CSS rate was observed in patients who received adjuvant chemotherapy than those who received surgery alone (5-year CSS: 64.1% vs. 61.1%, HR: 0.85, 95% CI: 0.79–0.93, *p* < 0.001). This improvement was significant in both the 65–80 (5-year CSS: 65.4% vs. 63.2%, HR: 0.91, 95% CI: 0.84–0.99, *p* = 0.030) and >80 (5-year CSS: 51.7% vs. 44.1%, HR: 0.52, 95% CI: 0.41–0.65, *p* < 0.001) subgroups. Figure 2 plots the OS and CSS curves according to the age of patients. Results of sensitivity analyses were consistent with those of the primary analysis, indicating that adjuvant chemotherapy improved OS (HR: 0.75, 95% CI: 0.71–0.79, *p* < 0.001) and CSS (HR: 0.83, 95% CI: 0.76–0.90, *p* < 0.001) as compared to surgery alone. Appendix A shows the results of the sensitivity analysis of OS and CSS.

Figure 3 shows the results of subgroup analysis of OS rates in patients who received either adjuvant chemotherapy or surgery alone. Adjuvant chemotherapy improved OS in the majority of the subgroups, regardless of age group, gender, any history of diabetes, serum albumin level, and eGFR level; however, patients with CCI ≥ 8 (HR: 1.17, 95% CI: 0.93–1.47, *p* = 0.168) did not benefit from adjuvant chemotherapy.

Among patients who received adjuvant chemotherapy, the incidence of G3/4 anemia was 30.9%, the incidence of neutropenia was 12.8%, and the incidence of thrombocytopenia 8.6%. The incidence of severe infection was 17.2% (febrile neutropenia 7.4%, pneumonia 7.5%, and urinary tract infection 2.3%). Table 2 shows the incidence of adverse events in patients with adjuvant chemotherapy. The incidence of early termination of chemotherapy was 19.1% (18.6% in patients aged 65–80 and 23.5% in patients aged >80).

Factors associated with a lower OS rate included no use of adjuvant chemotherapy (HR: 0.52, 95% CI: 0.41–0.66, *p* < 0.001), more advanced age (HR: 1.06, 95% CI: 1.04–1.09, *p* < 0.001), male sex (HR: 1.35, 95% CI: 1.04–1.75, *p* = 0.02), low serum albumin (<30 g/L) (HR: 1.47, 95% CI: 1.08–2.04, *p* = 0.017), and history of stroke (HR: 1.99, 95% CI: 1.01–3.93, *p* = 0.048). Appendix A shows risk factors associated with OS estimated via multivariable logistic regression models.

## 4. Discussion

This retrospective study demonstrated the efficacy of adjuvant chemotherapy in older patients with stage II/III gastric cancer. Several important findings in our study include the following points: (1) the overall use of adjuvant chemotherapy in older patients with gastric cancer was low, being around 35%; (2) adjuvant chemotherapy significantly improved the OS and CSS in older patients; (3) there was no significant difference in survival benefit between adjuvant monotherapy and double chemotherapy; (4) the survival benefit was seen in all subgroups, except those with a CCI ≥8; and (5) hematological adverse events were not uncommon.

Although previous RCTs demonstrated promising survival benefits related to adjuvant chemotherapy after curative gastrectomy, they failed to provide solid and consistent evidence to prove the efficacy of adjuvant chemotherapy in older patients with gastric cancer. In the CLASSIC trial, 1035 stage II-IIIB gastric cancer patients were randomised following D2 gastrectomy based on either adjuvant capecitabine/oxaliplatin (Capox) or surgery alone. Adjuvant Capox improved OS (5-year OS: 78% vs. 69%, *p* = 0.0015) and disease-free survival (DFS) (5-year DFS: 68% vs. 58%, *p* < 0.0001) compared to surgery alone [18]. Among patients aged ≥65 (n = 269, 26.0%), survival was also significantly improved via adjuvant chemotherapy (5-year OS: HR: 0.51; 95% CI 0.34–0.78; 3-year DFS: HR: 0.48; 95% CI 0.30–0.78). In the Japanese ACT-GS study, which randomised 1059 patients with stage II/III gastric cancer who received post-operative adjuvant chemotherapy with one year of S-1 (tegafur, gimeracil, and oteracil) or surgery alone, adjuvant S-1 improved the 5-year OS (71.7% vs. 61.1%, HR: 0.67, 95% CI 0.54–0.82) and DFS (65.4% vs. 53.1%, HR: 0.65, 95% CI 0.54–0.79) [19]. However, among the patients aged 70–80, the survival benefit could not be demonstrated (HR: 0.78; 95% CI 0.53–1.15). Likewise, there was no improvement in DFS among patients aged 60–69 (HR: 0.73; 95% CI 0.52–1.01) and 70–80 (HR: 0.71; 95% CI 0.49–1.02). A meta-analysis performed by Chang et al., which included data from the CLASSIC and ACTS-GS studies, confirmed that adjuvant chemotherapy had a significant impact on relapse-free survival (RFS) in older patients (HR: 0.61, 95% CI 0.47–0.81, *p* < 0.001), while the benefit on OS was marginal (HR: 0.75, 95% CI 0.55–1.01, *p* = 0.055) [21].

Previously, a few single-centre retrospective studies also focused on adjuvant chemotherapy in elderly patients after gastrectomy, though their results were inconsistent. Jeong et al.’s study included 130 patients aged ≥75 years old and showed no OS-related benefit of adjuvant chemotherapy (5-year OS: 44.1% vs. 30.7%, *p* = 0.804) [22]. Jin et al. reviewed 360 elderly Chinese patients who underwent D2 gastrectomy at a single institution [23]. Adjuvant chemotherapy significantly improved the OS rate in stage III patients (46.5 vs. 22.4 months, HR: 0.67, 95% CI 0.47–0.97, *p* = 0.033), but not in stage I or stage II patients (HR: 0.52, 95% CI 0.21–1.30, *p* = 0.161). Jo et al. reviewed 277 Korean patients aged ≥70 who underwent D2-gastrectomy and found that adjuvant chemotherapy significantly improved the RFS (35.5 vs. 20.4 months, HR: 0.50, 95% CI 0.27–0.96, *p* = 0.03), though no significant improvement in OS (*p* = 0.24) was recorded [24]. Liang et al.’s study included 270 elderly patients with stage II/III gastric cancer and demonstrated that adjuvant chemotherapy was significantly associated with better OS (HR: 0.57, 95% CI: 0.36–0.90, *p* = 0.017) and DFS (HR: 0.51, 95% CI: 0.32–0.81, *p* = 0.004) in stage III patients [25]. Nevertheless, the evidence identified in these studies may not be strong enough because they were conducted in single institutions, did not conduct matching with a control group, and had limited data regarding patients’ baseline comorbidities.

Our study fills the gap in our understanding of the use of adjuvant chemotherapy in a real-world setting. We conducted this retrospective study by querying the largest available cancer database, which included all cancer institutions in Hong Kong. We included a large number of elderly patients (n = 2616) with stage II/III gastric cancer who underwent D2-gastrectomy. Our study, which used PSM analysis, confirmed that adjuvant chemotherapy significantly improved both OS and CSS. An improvement in OS was also seen in those patients who were of advanced age. This information is important to both oncologists and patients, as many patients and families worry about toxicities of chemotherapy and prefer not to pursue adjuvant chemotherapy if toxicity is deemed a serious issue. The absolute values in median OS and 5-year OS rates were lower than those in the ACT-GS and CLASSIC studies, as the RCTs usually included younger and fit patients. Moreover, it is known that the OS rate of older patients with gastric cancer is shorter than those of younger patients. A study by Liang et al. demonstrated that patients aged ≥70 years had a significantly lower 5-year OS rate than the younger and middle-aged patients (values for elderly, middle-aged, and younger patients were 22.0%, 36.6%, and 38.0%, respectively) [26]. The worse prognosis in elder patients than in younger patients can be attributed to the delay in diagnosis and advanced tumour stage.

There have been frequent debates about whether to use monotherapy or doublet chemotherapies in the adjuvant setting. Previous meta-analyses and retrospective studies suggested that the adjuvant oxaliplatin–fluoropyrimidine combination might be more effective after curative resection, especially for patients with more advanced disease. It is certainly challenging for older patients to tolerate two chemotherapies that come with a higher risk of toxicity. While there were no significant differences between OS and CSS between monotherapy and doublet chemotherapy, in combination with higher toxicities with doublet chemotherapies, the use of adjuvant monotherapy with fluorouracil-based chemotherapy may be more suitable for older populations. There are ongoing RCTs comparing monotherapy and doublet agents in elderly patients, as well as the varied duration of adjuvant chemotherapy. The results of these studies are eagerly awaited.

What elderly patients concern most is probably not survival, but health-related quality of life and treatment toxicities. The incidences of grade 3/4 anemia and severe infection were higher than those captured in other RCTs. These adverse events would affect patients’ quality of life and their tolerability of subsequent chemotherapies. The rate of discontinuation of chemotherapy in our study was higher than that of other RCTs, e.g., the discontinuation rate was 10% in the CLASSIC trial [18]. Before starting chemotherapy, it is important for the oncologists to identify which patients may have a higher risk of severe toxicities. The CRASH and CARG toxicity tools are useful in predicting the risk of severe toxicities due to chemotherapy [27,28]. Moreover, patients and relatives should be provided with sufficient education on self-management of the common toxicities to reduce the chance of unexpected termination of treatment.

### Limitations

Our study was a real-world retrospective analysis that included a large number of elderly patients, and PSM was used to mitigate bias. There are several potential limitations. Since the pathology data and exact stage of the disease (i.e., T stage or N stage) were not available in the CMS database, separate efficacy assessments of stage II and III disease could not be performed. Data regarding non-hematological adverse events and patients’ quality of life, which are important concerns in older patients, were not included in the database. Patients in the surgery alone group were older and had substantially more comorbidities. To create matching groups, the oldest patients with multiple comorbidities were excluded after matching. Gastric cancer is more prevalent in the Asians, who are known to have better treatment outcomes than the Western patients. Moreover, adjuvant chemotherapy is often given to the Asian patients while peri-operative chemotherapy is more commonly used in the Western patients. It is unknown whether our results can be generalised to the Western populations.

## 5. Conclusions

In this real-world population-based study, we demonstrated the survival benefits of adjuvant chemotherapy among older patients with stage II/III gastric cancer after D2 gastrectomy. Future prospective studies are needed to provide personalised treatment for older populations, as they are highly variable in terms of their performance status, functional capacity, comorbidities, and social support. Health-related quality of life, which is a major concern of the older patients, should also be measured as one of the treatment outcomes in the prospective studies.

## Figures and Tables

**Figure 1 cancers-15-03768-f001:**
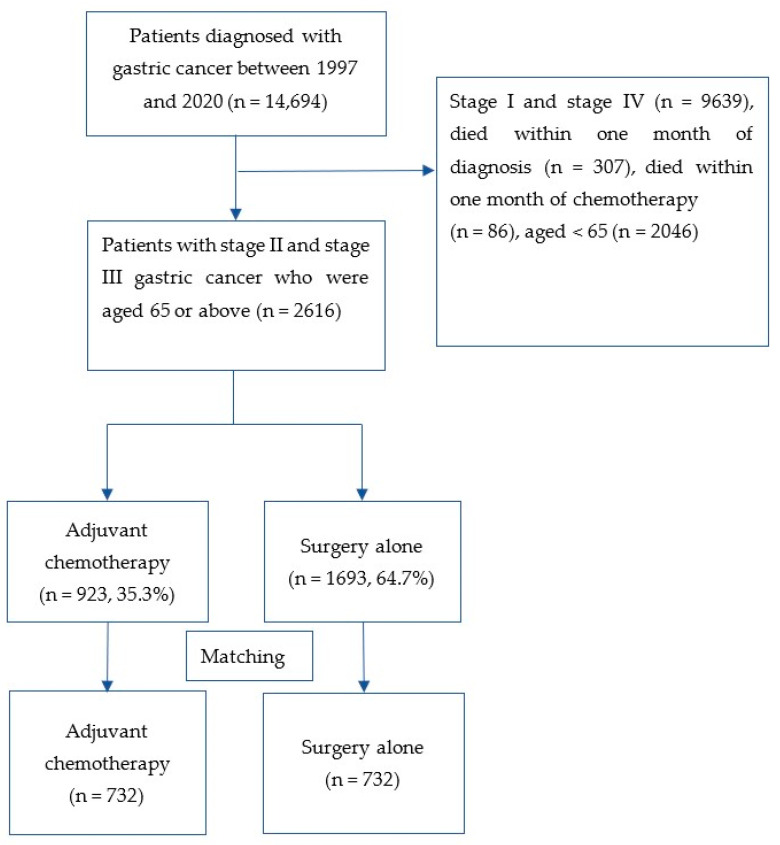
A flowchart that outlines the selection of patients who received adjuvant chemotherapy or surgery alone for gastric cancer.

**Figure 2 cancers-15-03768-f002:**
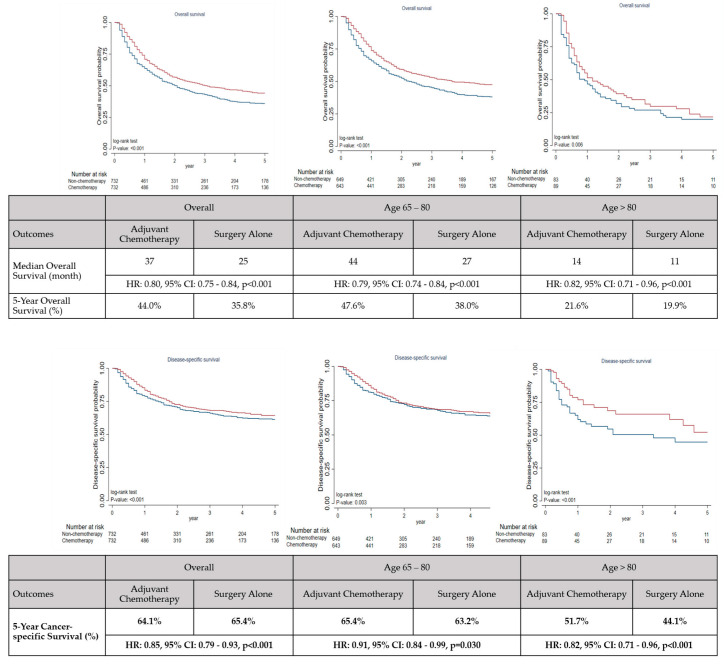
Kaplan–Meier curves of survival outcomes in older patients with gastric cancer who received either adjuvant chemotherapy or surgery alone.

**Figure 3 cancers-15-03768-f003:**
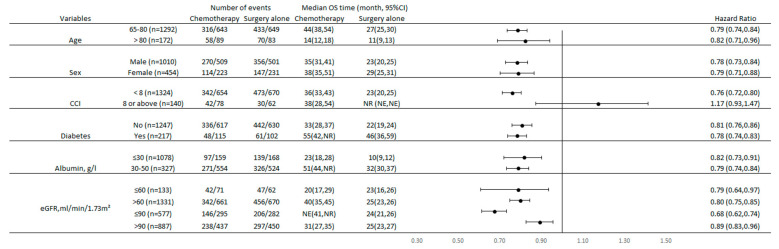
Subgroup analysis of overall survival in older patients with gastric cancer. After comparing monotherapy and doublet agents, we determined that the differences in OS (HR 1.00, 95% CI 0.91–1.09, *p* = 0.955) and CSS (HR 1.04, 95% CI 0.92–1.18, *p* = 0.502) were not statistically significant.

**Table 1 cancers-15-03768-t001:** Baseline characteristics of patients after propensity score matching.

	Total (n = 1464)	Non-Chemotherapy (n = 732)	Chemotherapy (n = 732)
**General information**			
Age (year), Mean ± SE	73.7 ± 0.2	74.4 ± 0.2	73.1 ± 0.2
Age group, %			
≤80	88.3	88.7	87.8
>80	11.8	11.3	12.2
Sex, n (%)			
Male	68.9	68.4	69.5
**Treatment**			
Radiotherapy, %	9.3	8.9	9.7
Chemotherapy drugs			
S-1	NA	NA	17.6
CAPOX	NA	NA	27.9
CAPECITABINE	NA	NA	45.6
FOLFOX4	NA	NA	4.5
OTHERS	NA	NA	4.4
**Chemotherapy regimen, %**			
Monotherapy	NA	NA	65.6
Doublet	NA	NA	34.4
**Comorbidity, %**			
Coronary heart disease	4.2	3.7	4.8
Heart failure	1.0	1.0	1.1
Stroke	3.4	3.1	3.6
Atrial fibrillation	1.4	1.4	1.5
Diabetes mellitus	18.5	20.1	16.9
Hypertension	24.4	24.3	24.5
Liver cirrhosis	1.8	1.5	2.1
CCI, Mean ± SE	6.2 ± 0.1	6.2 ± 0.1	6.1 ± 0.1
**CCI, n (%)**			
≤7	90.4	91.5	89.3
Eight or above	9.6	8.5	10.7
**Biological parameters, Mean ± SD**			
WBC, 10^9^/L	8.1 ± 0.1	8.1 ± 0.1	8.1 ± 0.1
RBC, 10^12^/L	3.8 ± 0.1	3.8 ± 0.1	3.9 ± 0.1
Platelet, 10^9^/L	277.4 ± 2.7	275.8 ± 3.7	279.7 ± 3.9
Neutrophil, 10^9^/L	6.1 ± 0.1	6.3 ± 0.1	6.0 ± 0.2
Lymphocyte, 10^9^/L	1.4 ± 0.2	1.3 ± 0.2	1.4 ± 0.3
ALP, U/L	80.9 ± 1.7	84.6 ± 2.8	77.2 ± 1.9
Total bilirubin, μmol/L	10.4 ± 0.2	10.7 ± 0.1	10.0 ± 0.2
eGFR, mL/min/1.73 m^2^	103.0 ± 0.8	103.9 ± 1.3	102.1 ± 1.1
Albumin, g/L	33.9 ± 0.2	34.0 ± 0.2	33.8 ± 0.2

Abbreviation: SD = standard deviation; CCI = Charlson Comorbidity Index; WBC = white blood cell; Hb = hemoglobulin level; ALP = alkaline phosphatase; CrCl = creatinine clearance; eGFR = estimated glomerular filtration rate; NA = not available. The calculation of the Charlson Comorbidity Index does not include Acquired Immune Deficiency Syndrome (AIDS).

**Table 2 cancers-15-03768-t002:** Incidence of hematological toxicities in older patients who received adjuvant chemotherapy.

Grade 3/4 Hematological Toxicities	Percentage
Anemia	30.9%
Neutropenia	12.8%
Thrombocytopenia	8.6%
Febrile neutropenia	7.4%
Pneumonia	7.5%
Urinary tract infection	2.3%

## Data Availability

The data presented in this study are available on request from the corresponding author. The data are not publicly available due to concerns related to patients’ privacy.

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
