# Peer review of "Adjuvant Chemotherapy in Older Patients with Gastric Cancer: A Population-Based Cohort Study"

_cancers, 2023, doi:10.3390/cancers15153768_

Round 1
Reviewer 1 Report
Very nice retrospective propensity matched study.
1) Control group surgery alone worse than previously reported studies. Please explain.
2) CSS very minimal in all groups except greater than 80 . PLease explain.
3) In the older greater than 80 group what were the causes of OS. Thank you.
Author Response
Very nice retrospective propensity matched study.
1) Control group surgery alone worse than previously reported studies. Please explain.
Thank you very much for your comment. The 5-year overall survival of our patients with surgery alone was 35.8%. The overall survival was shorter than in other randomized controlled trials. For example, in the CLASSIC trial, the 5-year OS was 78% in adjuvant chemotherapy arm and 69% in the control arm. Patients in randomised controlled trials usually have better general condition and have a longer survival. When compared with the survival in studies on older patients with cancer, the 5-year overall survival in our study was similar. A study by Liang et al. demonstrated that patients aged ≥ 70 years had a significantly lower 5-year OS rate than the younger and middle-aged patients (elderly vs middle-aged vs younger patients, 22.0% vs 36.6% vs 38.0%, respectively). The worse prognosis in elder patients can be attributed to the delay in diagnosis and advanced tumor stage.
This has been added in the discussion part.
2) CSS very minimal in all groups except greater than 80 . PLease explain.
The cancer-specific survival was significantly improved after chemotherapy in all subgroups. Since majority of the patients in our study were aged 60-79 and the benefit of chemotherapy in cancer-specific survival was not so pronounced in this age group, the overall cancer-specific survival was marginally statistical significant.
Moreover, for patients over 80 years old, their comorbidities may preclude them from having operation or chemotherapy. If the patient can have surgery, their general conditions are usually better than those who cannot go for surgery. Their prognosis and survivals will probably be affected by the cancer recurrence rather than due to other diseases. This can also explain why CSS is more prominent in patients over 80 years old.
3) In the older greater than 80 group what were the causes of OS. Thank you.
Thank you very much for your question. There are multiple causes of death in patients over 80 years old, including ischemic heart disease, respiratory disease, infection, cancer recurrence, liver failure, chronic renal disease, etc.
Reviewer 2 Report
The authors evaluated the role of advent chemotherapy in locally advanced gastric cancer stage II/III after radical gastrectomy with D2 lymphadenectomy. They found that patients after adjuvant treatment had better OS and CSS. It is interesting result especially in older population where compliance with chemotherapy may be unsatisfactory. The great number of patients is a strong advantage of the paper. Some crucial issues should be addressed before possible publication.
- The baseline characteristics of the patients (Table 1) should include precise information on pT categories (number of patients with pT1, pT2, pT3, pT4 cancers) and pN status. Moreover exact number of patients with stage II and stage III disease should be specified.
- When survival analysis is performed in propensity score matching stage of cancer should be included. There is no information in Table 1 whether patients in both analysed groups have comparable stage of cancer - how many patients with cancer stage II and III were in analysed groups? More important than biological agents used for PSM is including of cancer stage into PSM.
- Compliance to chemotherapy should be mentioned or discussed. As in the elderly the compliance is probably worse than in younger population. What about dose reduction during chemotherapy?
- Biological parameters were used for baseline characteristics and PSM, when were they collected? Before surgery? Before starting chemotherapy?
- PSM was performed and patients were matched for age, gender, comorbidities and biological parameters. What calliper was used for PSM? Why patients with comparable age and comorbidities did not receive adjuvant chemotherapy but only surgery? What rationale was behind the decision that some patients underwent surgery only as combined treatment was a standard approach? This is important as CCI was comparable between matched groups.
- In the results section there is a paragraph not associated with the manuscript: “This section may be divided by subheadings. It should provide a concise and precise 141 description of the experimental results, their interpretation, as well as the experimental 142 conclusions that can be drawn. “
- The title of Table 1 is “Baseline characteristics of patients before propensity score matching “ whereas it present data after PSM.
- 9,3% of patients received radiotherapy. What were the indications for postoperative radiotherapy? Which chemotherapy regimens were used together with radiotherapy? If the exact data is unavailable the issue should be commented or briefly discussed. Maybe patients with radiotherapy should be excluded from analyses to eliminate potential bias? There is high number of patients in both groups so excluding radiotherapy patients will not deteriorate the value of the paper.
- In Supplementary Table 2 most important factors influencing survival are not taken into consideration: pT stage and pN stage should be added to multivariate analysis.
Author Response
The authors evaluated the role of advent chemotherapy in locally advanced gastric cancer stage II/III after radical gastrectomy with D2 lymphadenectomy. They found that patients after adjuvant treatment had better OS and CSS. It is interesting result especially in older population where compliance with chemotherapy may be unsatisfactory. The great number of patients is a strong advantage of the paper. Some crucial issues should be addressed before possible publication.
- The baseline characteristics of the patients (Table 1) should include precise information on pT categories (number of patients with pT1, pT2, pT3, pT4 cancers) and pN status. Moreover exact number of patients with stage II and stage III disease should be specified.
Thank you very much for your comment. This is actually a limitation of our study and we have included this in our discussion part (limitation). Since the Hospital Authority CMS database did not have the records of patients’ T and N stage or exact Stage II/ III disease, the exact number of patients in stage II/ III cannot be sorted out. More, we cannot do separate efficacy assessments according to stage.
- When survival analysis is performed in propensity score matching stage of cancer should be included. There is no information in Table 1 whether patients in both analysed groups have comparable stage of cancer - how many patients with cancer stage II and III were in analysed groups? More important than biological agents used for PSM is including of cancer stage into PSM.
Thank you very much for your comment. Similar to question 1, the Hospital Authority CMS did not have the records of patients’ exact T / N stage. Therefore the chemotherapy used for different stages of the disease could not be retrieved.
- Compliance to chemotherapy should be mentioned or discussed. As in the elderly the compliance is probably worse than in younger population. What about dose reduction during chemotherapy?
Thank you very much for your question. We have calculated the early termination rate and have included this in the result part. Early termination of chemotherapy was seen in 19.8% in patients aged 65-80 and 23.2% in patients aged > 80.
- Biological parameters were used for baseline characteristics and PSM, when were they collected? Before surgery? Before starting chemotherapy?
Thank you very much for your question. This is an important piece of information and we have missed. The baseline characteristics and biological parameters were collected after surgery and before starting adjuvant chemotherapy. We have edited this in the “method” part.
- PSM was performed and patients were matched for age, gender, comorbidities and biological parameters. What calliper was used for PSM? Why patients with comparable age and comorbidities did not receive adjuvant chemotherapy but only surgery? What rationale was behind the decision that some patients underwent surgery only as combined treatment was a standard approach? This is important as CCI was comparable between matched groups.
Thank you very much for your question. The decision to receive adjuvant chemotherapy depends on many factors, including patients’ willingness to receive chemotherapy, patients and families’ acceptance, physicians’ choices, availability of the patient to see oncologists, etc. In the Chinese population, many decisions are not made by the elderly patients. Rather, most of the time, family members would make the final decision for the patients. Many patients have the misconception that chemotherapy will harm the patients and will have lots of intolerable side effects. That’s why they decided not to receive adjuvant chemotherapy.
- In the results section there is a paragraph not associated with the manuscript: “This section may be divided by subheadings. It should provide a concise and precise 141 description of the experimental results, their interpretation, as well as the experimental 142 conclusions that can be drawn. “
I am sorry for that. That’s the original template from “Cancers” and I forgot to delete. I am sorry for the confusion.
- The title of Table 1 is “Baseline characteristics of patients before propensity score matching “ whereas it present data after PSM.
Sorry for the typo. I have edited this already.
- 9,3% of patients received radiotherapy. What were the indications for postoperative radiotherapy? Which chemotherapy regimens were used together with radiotherapy? If the exact data is unavailable the issue should be commented or briefly discussed. Maybe patients with radiotherapy should be excluded from analyses to eliminate potential bias? There is high number of patients in both groups so excluding radiotherapy patients will not deteriorate the value of the paper.
Thank you very much for your comment. The indications for adjuvant radiotherapy included patients with close or positive resection margin. Since there was only a small proportion of patients receive adjuvant radiotherapy, our team decided to keep the current data instead of excluding them.
- In Supplementary Table 2 most important factors influencing survival are not taken into consideration: pT stage and pN stage should be added to multivariate analysis.
Thank you very much for your comment. This is actually a limitation of our study and we have included this in our discussion part (limitation). Since the Hospital Authority CMS database did not have the records of patients’ T and N stage or exact Stage II/ III disease, we cannot do multivariate analysis with inclusion of T and N stages.